# Pharmacokinetics of oral moxidectin in individuals with *Onchocerca volvulus* infection

Beesan Tan[1], Nicholas Opoku[2,3], Simon K. Attah[2], Kwablah Awadzi[2†], Annette C. Kuesel[4], Janis Lazdins-Helds[4], Craig Rayner[5], Victoria Ryg-Cornejo[6], Mark Sullivan[6,7], Lawrence Fleckenstein[1]*

1 College of Pharmacy, University of Iowa, Iowa City, Iowa, United States of America, 2 Onchocerciasis Chemotherapy Research Centre, Hohoe, Ghana, 3 University of Health and Allied Sciences (UHAS), Ho, Ghana, 4 UNICEF/UNDP/World Bank/World Health Organization Special Programme for Research and Training in Tropical Diseases (WHO/TDR), World Health Organization, Geneva, Switzerland, 5 Monash University, Victoria, Australia, 6 Medicines Development for Global Health, Southbank, Victoria, Australia, 7 The Kirby Institute, University of New South Wales, Sydney, New South Wales, Australia

† Deceased.
* l-fleckenstein@uiowa.edu

**Data Availability Statement:** Individual participant data cannot be made openly accessible due to ethical and privacy considerations, as reflected in the information document for the participants and

## Abstract

### Background

Onchocerciasis ("river blindness"), is a neglected tropical disease caused by the filarial nematode *Onchocerca volvulus* and transmitted to humans through repeated bites by infective blackflies of the genus *Simulium*. Moxidectin was approved by the United States Food and Drug Administration in 2018 for the treatment of onchocerciasis in people at least 12 years of age. The pharmacokinetics of orally administered moxidectin in 18- to 60-year-old men and women infected with *Onchocerca volvulus* were investigated in a single-center, ivermectin-controlled, double-blind, randomized, single-ascending-dose, ascending severity of infection study in Ghana.

### Methodology/Principal findings

Participants were randomized to either a single dose of 2, 4 or 8 mg moxidectin or ivermectin. Pharmacokinetic samples were collected prior to dosing and at intervals up to 12 months post-dose from 33 and 34 individuals treated with 2 and 4 mg moxidectin, respectively and up to 18 months post-dose from 31 individuals treated with 8 mg moxidectin. Moxidectin plasma concentrations were determined using high-performance liquid chromatography with fluorescence detection. Moxidectin plasma $AUC_{0-\infty}$ (2 mg: 26.7–31.7 days*ng/mL, 4 mg: 39.1–60.0 days*ng/mL, 8 mg: 99.5–129.0 days*ng/mL) and $C_{max}$ (2mg, 16.2 to17.3 ng/mL, 4 mg: 33.4 to 35.0 ng/mL, 8 mg: 55.7 to 74.4 ng/mL) were dose-proportional and independent of severity of infection. Maximum plasma concentrations were achieved 4 hours after drug administration. The mean terminal half-lives of moxidectin were 20.6, 17.7, and 23.3 days at the 2, 4 and 8 mg dose levels, respectively.

### Conclusion/Significance

We found no relationship between severity of infection (mild, moderate or severe) and exposure parameters ($AUC_{0-\infty}$ and $C_{max}$), $T_{1/2}$ and $T_{max}$ for moxidectin. $T_{max}$, volume of

informed consent the participants gave. Anonymized data may be made available on a case to case basis and researchers should contact Sally Kinrade (email: sally.kinrade@medicinesdevelopment.com) at the following address: Medicines Development for Global Health, 1/18 Kavanagh Street, Southbank VIC 3004, Australia. All raw and composite data can be found at this address.

**Funding:** WHO (UNICEF/UNDP/World Bank/WHO Special Programme for Research and Training in Tropical Diseases, TDR) funded this study, utilizing contributions from the WHO African Programme for Onchocerciasis Control (APOC) and WHO/TDR donor countries. Grant funding provided to the institution of KA included the salaries of Drs KA and NO. WHO provided grant funding for the determination of moxidectin plasma concentrations and the pharmacokinetic analysis to the institution of Drs LF and BT. Wyeth provided the study drugs, contributed to the study design and protocol, prepared the regulatory submission and provided data management services. Following Pfizer take-over of Wyeth, Pfizer continued to provide data management services, but did not contribute to the analyses, this manuscript or the decision to publish.

**Competing interests:** I have read the journal's policy and the authors of this manuscript have the following competing interests: This study was funded by the UNICEF/UNDP/World Bank/World Health Organisation Special Programme for Research and Training in Tropical Diseases (WHO/TDR) and the African Programme for Onchocerciasis Control. JLH and ACK were staff of WHO/TDR at the time of study preparation and conduct, were responsible for the management of the project at WHO/TDR and confirm that their employment and role in project management has not caused any conflict of interest.

distribution (V/F) and oral clearance (CL/F) are similar to those in healthy volunteers from Europe. From a pharmacokinetic perspective, moxidectin is an attractive long-acting therapeutic option for the treatment of human onchocerciasis.

## Author summary

The 2017 Global Burden of Disease Study estimated 20.9 million individuals with onchocerciasis, primarily in Africa. Onchocercal vision impairment/blindness and skin disease (e.g., skin pigment loss, debilitating itching) impact the social and economic life of infected individuals and their communities. This motivates onchocerciasis elimination efforts, today primarily through annual or biannual ivermectin treatment of affected communities. Despite progress towards elimination in many areas, others are not progressing well towards elimination and may require alternative treatment strategies. Moxidectin, approved by the United States Food and Drug Administration in 2018 for treatment of onchocerciasis in people at least 12 years old, could be an alternative. How the amount of a drug in the body changes over time is important for choosing a dose and treatment regimen and for regulatory approval. We measured moxidectin blood levels in 18 to 60 year old men and women with onchocerciasis. We found that moxidectin blood levels peaked around three-four hours after ingestion, that moxidectin stayed in the body for a long time (i.e., its elimination half-life was around 20 days) and that moxidectin blood levels depended on the dose, but not the infection severity as measured by the number of onchocerciasis parasites in the skin.

## Introduction

Human onchocerciasis, commonly known as "river blindness", is a neglected tropical disease caused by the filarial nematode *Onchocerca volvulus* and transmitted to humans through exposure to repeated bites of infective blackflies of the genus *Simulium*. Severe visual impairment and ultimately blindness are the best known and most serious morbidity, but onchocerciasis also causes skin lesions, depigmentation (leopard skin), and debilitating severe itching. The socio-economic burden of visual impairment/blindness and skin disease has motivated large scale disease control and elimination efforts [1,2]. The Global Burden of Disease Study estimated in 2017 that there were 20.9 million *O. volvulus* infected individuals worldwide, with 99% of the cases in Sub-Saharan Africa [3]. The disease is also endemic in Yemen and in the Amazon border region between Venezuela and Brazil [4].

Humans are the definitive host of the *O. volvulus* parasite. Adult male and female worms, known as macrofilariae, produce the progeny stage, known as microfilariae (mf). The immune response to dying and dead microfilariae in the skin and the eyes is primarily responsible for the pathology associated with onchocerciasis. Blackflies ingest skin-dwelling microfilariae when taking a blood meal from infected humans, initiating the vector stage of the life cycle. In the blackfly, microfilariae develop into infective (L3) larvae in about a week. During a subsequent blood meal from a human, the L3 larvae are released from the fly's mouthparts into the wound created by the feeding blackfly, developing into adult worms in about 10–15 months to complete the cycle [1,5].

Onchocerciasis control has now for many years been based on annual and in some areas biannual mass drug administration of ivermectin, a broad-spectrum antiparasitic macrocyclic lactone. Ivermectin's effects on *O. volvulus* include a microfilaricidal (killing of microfilariae) and an embryostatic (temporary inhibition of microfilarial release from female worms) effect. Adult

female worms gradually resume the production and release of microfilariae weeks to months after ivermectin treatment, requiring ongoing re-treatment. Repeated doses of ivermectin are postulated from modeling data to have a cumulative sterilizing effect on adult worms with an estimated 30–35% reduction in production of microfilariae per dose, and may lead to reduced female macrofilariae life expectancy (macrofilaricidal (killing of macrofilariae) effect) [6,7,8].

Despite progress towards elimination in many endemic areas in Africa, others are not on track to achieve elimination [9,10]. Some areas have reported sub-optimal responses to ivermectin raising concerns about reaching elimination goals [11–14]. This has led to renewed efforts to identify alternative treatment strategies aimed at accelerating progress toward elimination of transmission [15,16]. These strategies include broadening geographical coverage into hypoendemic areas, enhancing treatment coverage and frequency, exploring complementary vector control approaches and identifying new therapeutics.

Moxidectin was approved by the United States Food and Drug Administration in 2018 for the treatment of onchocerciasis in people at least 12 years of age (https://www.accessdata.fda.gov/drugsatfda_docs/label/2018/210867lbl.pdf). Like ivermectin, moxidectin is a macrocyclic lactone but of the milbemycin, not the avermectin, class and is a semisynthetic compound derived from a fermentation product of the actinomycete, *Streptomyces cyanogriseus*. Moxidectin is more lipophilic (log P = 5.4) than ivermectin (log P = 4.3), resulting in a larger volume of distribution, greater uptake in adipose tissue, and partially explaining moxidectin's longer half-life [17]. Clinical studies in *O. volvulus* infected individuals completed in Ghana, Liberia and the Democratic Republic of the Congo demonstrated moxidectin's superiority compared to ivermectin in terms of the extent and duration of reduction of skin microfilariae density, indicating its potential to accelerate and enhance the feasibility of onchocerciasis elimination [18,19]. The safety profile of moxidectin was similar to that of ivermectin. The adverse events reported in moxidectin and in ivermectin treated participants were mostly related to the inflammatory reactions to dying/dead microfilariae, known as Mazzotti reactions. Symptoms included itching, rashes, muscle pains, fever, tender lymph nodes, hypotension, tachycardia, and eosinophilia. Mazzotti reactions were seen more frequently in the moxidectin- than ivermectin-treated participants, likely related to a more pronounced killing of microfilariae. Adverse events were mild or moderate, did not require medical intervention and were self-limiting in nature [18,19]. For WHO and endemic countries to consider use of moxidectin for onchocerciasis control and elimination, additional safety data, including in individuals without detectable levels of skin microfilariae, efficacy data after multiple annual or biannual moxidectin treatment and a dose for children less than 12 years are needed and the relevant studies are currently ongoing (https://clinicaltrials.gov/ct2/show/NCT04311671, https://clinicaltrials.gov/ct2/show/NCT03876262, https://clinicaltrials.gov/ct2/show/NCT03962062). Furthermore, for decisions on any use in onchocerciasis–loiasis co-endemic areas, data are needed on the safety of moxidectin at different levels of *Loa loa* microfilaraemia.

Pharmacokinetics is an essential component of drug development and plays a vital role in establishing the optimal and effective use for a new drug. The aim of this work is to report on the pharmacokinetic characteristics of moxidectin in *O. volvulus* infected African men and women determined during a single-ascending-dose, ascending severity of infection Phase 2 study conducted between 2006 and 2009 [18].

## Methods

### Ethics statement

The study was conducted according to the principles of the Declaration of Helsinki and in compliance with Good Clinical Practice. Participants gave informed consent to study

participation and testified to this by signature or thumbprint in the presence of an independent literate witness in their villages before initiation of any study related procedures. The study was approved by the Ghana Food and Drugs Board, the Ghana Health Service Ethics Review Committee and the WHO Ethics Review Committee.

### Recruitment area

Participants were recruited from villages in the Tordzi basin in the Volta Region of South-eastern Ghana. The area was not included in vector control activities conducted by the Onchocerciasis Control Programme in West Africa because it was forested and not included in mass ivermectin distribution programmes because it was onchocerciasis hypoendemic (details given in [18]).

### Clinical study design and methodology

This was a randomized, single-ascending-dose, ivermectin-controlled, double-blind, safety and tolerability study of orally administered moxidectin in *Onchocerca volvulus* infected individuals. Secondary objectives included assessment of the pharmacokinetics and the effect of moxidectin on the parasites. A detailed description of study design and rationale as well as the methodology has been provided by Awadzi et al [18].

Briefly, participants were enrolled in 9 consecutive cohorts and randomized in each cohort by sex in a ratio of 3:1 to receive a single oral dose of moxidectin (2, 4, or 8 mg, representing 27–187 µg/kg or 0.04–0.29 µmol/kg) or the dose of ivermectin used by onchocerciasis control programs (150 µg/kg with 3 mg tablets). Based on consideration of participant safety, each moxidectin dose level was evaluated sequentially in three cohorts of participants with different severity of infection, defined by level of skin mf density and ocular involvement pre-treatment. In cohort 1, 4 and 7, participants had a skin mf density < 10 mf/mg skin and no ocular involvement ("mildly infected"). Participants in cohorts 2, 5 and 8 had a skin mf density of 10–20 mf/mg skin and the sum of mf in both eyes was ≤ 10 ("moderately infected"). Cohorts 3, 6 and 9 included participants with skin mf density >20 mf/mg skin with or without ocular involvement ("severely infected") and enrolled approximately twice as many participants as the mildly and moderately infected cohorts but blood samples for pharmacokinetics were obtained from only half of these. Treatment was taken under observation after an overnight fast.

Participants had to be healthy as determined by physical examination, electrocardiography, medical and medication history, serum biochemistry, hematology and semi quantitative urinalysis, without history of or current neurological or neuropsychiatric disease or epilepsy, orthostatic hypotension at screening, or hyperreactive onchodermatitis. Individuals who had received antifilarial therapy within the previous 5 years, were pregnant or breastfeeding, had a history of drug or alcohol abuse or reported regular use of ≥ 3 cigarettes per day were excluded. Women of child-bearing potential were on contraception (depo-medroxyprogesterone acetate or levonorgestrel implants) during the first 150 days after treatment. Use of alcohol or other drugs of abuse within 72 hours before study drug administration was not allowed.

### Blood sampling for determination of moxidectin plasma concentrations

Whole blood (6 mL) was collected into single evacuated blood tubes containing lithium heparin within 2 hours prior to dosing (0 h), at 1, 2, 4, 8, 24, and 72 hours after study drug administration, on days 8, 13, 18, and months 1, 2, 3, 6, and 12 post-dose from all participants in the cohorts evaluating 2 or 4 mg moxidectin and in addition 18 months post-dose from participants in the cohorts evaluating 8 mg moxidectin.

The blood collection tube was placed immediately on wet ice and centrifuged at $1500 \times g$ for 10 minutes in a refrigerated centrifuge within 15 minutes after collection. The plasma was harvested and distributed evenly into two labeled polypropylene containers. The aliquots were stored frozen at approximately -80˚C until shipment of one aliquot for analysis at the Clinical Pharmacokinetics Laboratory, University of Iowa, Iowa City, Iowa, USA. The samples were shipped in watertight sealed containers in insulated shipping containers packed with dry ice by air express service along with an inventory of the specimens identifying samples by participant identifier, sampling date and time of sampling relative to study drug administration. Upon receipt by the Clinical Pharmacokinetics Laboratory, the samples were inventoried and logged into the -80˚C freezer and stored frozen until analysis.

## Bioanalytical methodology

Moxidectin plasma concentrations were determined using high-performance liquid chromatography with fluorescence detection, as previously described with minor modifications as used in previous Phase 1 studies [20]. Briefly, the method used solid-phase extraction and fluorescent derivatization with trifluoroacetic anhydride and N-methylimidazole. Separation was achieved on a Symmetry $C_{18}$ column, 5 μm particle size, 250 mm × 4.6 mm (Waters, Milford, MA), with a mobile phase of tetrahydrafuran-acetonitrile-water (40,38,22 v/v/v). In this study, 500 μL of plasma was extracted and the range of quantitation was 0.08–120 ng/mL. The method validation studies had shown that the mean recovery of moxidectin at concentrations of 0.2–1000 ng/mL was 84.6%, the interday accuracy (mean bias) was <6.4% and the precision was 13.1% for moxidectin in plasma. Moxidectin stability was demonstrated for plasma samples left on the bench-top for 96 hours, through 3 freeze-thaw cycles and during storage for 2 weeks at +4˚C and for up to 6 months at −80˚C. All plasma samples were assayed within 4 months of sample collection.

## Pharmacokinetic and statistical analyses

Pharmacokinetic parameters were calculated by noncompartmental analysis using Win-Nonlin Professional (version 5.0, Pharsight Corporation, Mountain View, CA). For the analysis, concentrations below lower limit of quantification were treated as missing. Peak plasma concentration ($C_{max}$) and time to peak concentration ($T_{max}$) were determined for each individual without interpolation. The log-linear portion of the last 3–6 points of each set of plasma concentrations was used to determine the terminal phase hybrid constant, λz.

$AUC_{0-\infty}$ values were the sum of $AUC_{last}$ and AUC extrapolated to infinity. The $AUC_{last}$ was calculated by summation of each individual area between two consecutive time intervals from time 0 to the last measurable concentration ($C_{last}$) using the linear trapezoidal rule. AUC extrapolated was calculated by dividing $C_{last}$ by λz. Plasma half-life was calculated as: $T_{1/2} = \ln(2)/\lambda z$; total body apparent clearance for oral administration CL/F = Dose/ $AUC_{0-\infty}$; and apparent volume of distribution: V/F = Dose / (λz $^*AUC_{0-\infty}$).

Dose-proportionality of moxidectin pharmacokinetics and the effect of infection severity on moxidectin pharmacokinetics were assessed using analysis of variance (ANOVA) method. The analysis was performed on dose-normalized and natural log-transformed moxidectin exposure parameters ($AUC_{0-\infty}$ and $C_{max}$), using dose, infection severity and interaction term between dose and infection severity as fixed factors. **P**-values of <0.05 were deemed to indicate statistical significance. Summary statistics of the pharmacokinetic parameters and the ANOVA analysis were conducted using R software (version 3.2.2, R Core Team).

## Results

### Demographic data

Baseline demographics for the 98 participants from whom pharmacokinetic samples were obtained were similar across all treatment groups. Mean age was 38.3 years. (range 18–58), 37.6 (19–57) and 32.7 (18–60) for the 2, 4 and 8 mg dose groups, respectively. Mean weight was 59.9 kg (range 45.8–86.9), 57.6 kg (43.3–74.0) and 59.5 kg (range 42.7–88.7) for the 2, 4 and 8 mg dose groups, respectively. The number of women was 7 of 33, 12 of 34 and 7 of 31 for the 2, 4 and 8 mg dose groups, respectively. All participants were Africans. Table 1 shows severity of infection by moxidectin dose and cohort.

### Pharmacokinetic parameters

Exposure to moxidectin in terms of $AUC_{0-\infty}$ and $C_{max}$ was dose proportional for the doses studied (2, 4, and 8 mg) following single-dose administration. Fig 1 shows the geometric mean plasma concentration-time profiles of moxidectin for the participants who received 2, 4 and 8 mg of moxidectin using a linear scale, illustrating dose proportionality. Table 2 shows the pharmacokinetic parameters of moxidectin for the three dose levels studied across all participants without regard to severity of infection. The median $T_{max}$ was 4 hours. The arithmetic mean $T_{1/2}$ ranged from 17.7 to 23.3 days. Pharmacokinetic exposure parameters ($AUC_{0-\infty}$ and $C_{max}$), $T_{1/2}$ and $T_{max}$ were not affected by severity of infection (Table 3). Moxidectin is a low clearance drug with a mean oral clearance (CL/F) ranging from 80.3 to 93.9 L/day and a relatively apparent large volume of distribution (V/F), with mean values ranging from 2222 to 2421 L and not affected by severity of infection (Table 3). Figs 2–4 illustrate the lack of effect of

**Table 1. Moxidectin dose and severity of infection by cohort.**

| Cohort No. (N)[1] | Moxidectin Dose[2] | Severity of Infection[3] | Microfilariae/mg skin[4] Mean (SD) |
|---|---|---|---|
| 1 (10) | 2 mg | Mild | 2.7 (2.5) |
| 2 (11) | | Moderate | 15.4 (2.8) |
| 3 (12) | | Severe | 39.6 (7.8) |
| 4 (11) | 4 mg | Mild | 3.8 (2.3) |
| 5 (11) | | Moderate | 13.1 (2.2) |
| 6 (12) | | Severe | 31.1 (7.8) |
| 7 (12) | 8 mg | Mild | 4.4 (2.8) |
| 8 (11) | | Moderate | 13.5 (2.7) |
| 9 (8) | | Severe | 42.5 (17.8) |

[1]Cohorts screened for and eligible participants randomized and treated in sequential order. N is the number of moxidectin treated participants from whom blood samples for pharmacokinetic analysis were obtained.

[2]In each dose group, participants were randomized 3:1 to the dose of moxidectin specified and ivermectin, respectively.

[3]Severity of infection defined in text.

[4]Arithmetic mean microfilariae/mg skin (SD, standard deviation) across all participants in the mildly, moderately, and severely infected cohort included in the pharmacokinetic analysis were 3.7 ± 2.6, 14.0 ± 2.7 and 37.1 ± 15.1 mf/mg skin.

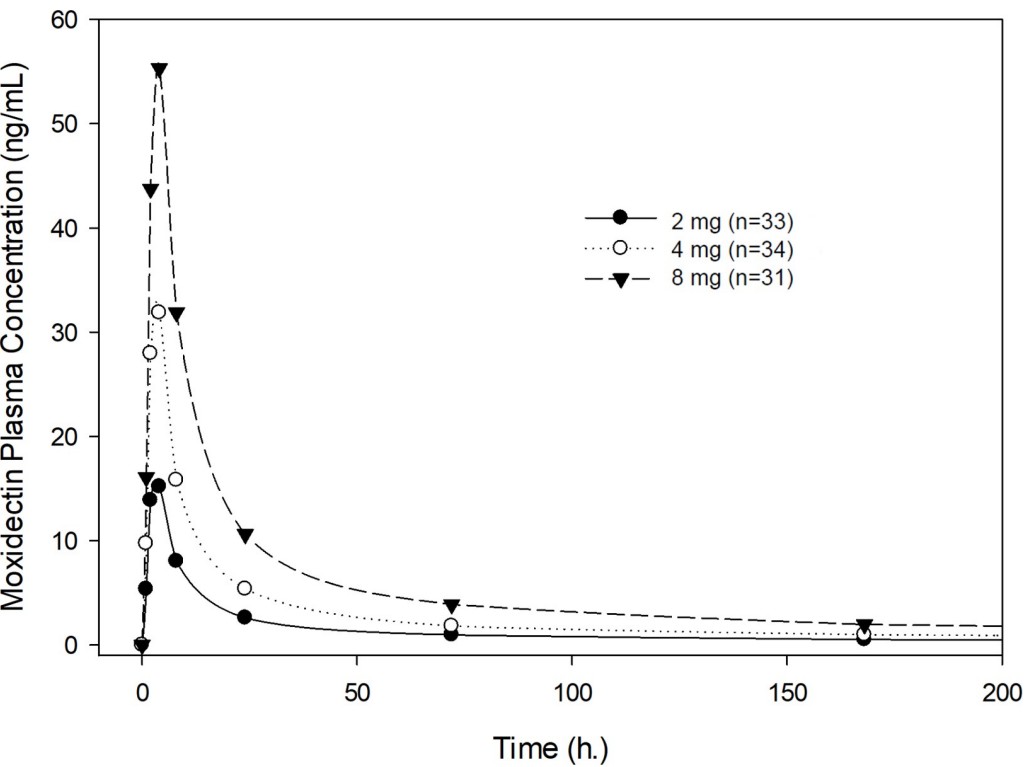

**Fig 1. Plasma concentrations of moxidectin after single oral administration of 2, 4 or 8 mg, displayed as geometric means.**

severity of infection on exposure. The $C_{max}$ in Fig 4 is higher for the severely than the mildly or moderately infected cohort, but this is considered due to the rapid and pronounced distribution phase of moxidectin in conjunction with the fact that sampling was only hourly near the peak concentration. Dose-normalized $AUC_{0-\infty}$ and $C_{max}$ are presented as box-whisker plots by dose and severity of infection in Fig 5, showing similar distribution of the exposure parameters across different moxidectin dose levels (2–8 mg) and different severities of infection. As illustrated in Table 4, the dose-normalized moxidectin exposures were not statistically different across all treatments and severity of infection when analyzed with the ANOVA model.

## Discussion

We described herein the pharmacokinetics of moxidectin in *O. volvulus* infected Africans who received a single oral moxidectin dose of 2, 4, or 8 mg. The plasma drug concentration profile

**Table 2. Summary of PK parameters of moxidectin for dose levels studied.**

| Dose (mg) | PK Parameter[a] | | | | | |
|---|---|---|---|---|---|---|
| | $C_{max}$ (ng/mL) | $T_{max}$ (h) | $AUC_{0-\infty}$ (days*ng/mL) | $T_{1/2}$ (days) | CL/F (L/day) | V/F (L) |
| 2 (n = 33) | 16.6 (4.1) | 4.0 (2.0, 4.2) | 28.8 (12.5) | 20.6 (10.4) | 80.3 (28.5) | 2222 (970) |
| 4 (n = 34) | 34.3 (6.6) | 4.0 (2.0, 4.2) | 48.7 (20.3) | 17.7 (8.6) | 93.9 (34.3) | 2224 (988) |
| 8 (n = 31) | 63.1 (20.0) | 4.0 (1.2, 4.0) | 114.1 (66.9) | 23.3 (21.9) | 84.0 (29.4) | 2421 (1658) |

[a]Values are given as arithmetic means (SD) except for $T_{max}$, which shows median and (min, max).

**Table 3. Summary of PK parameters of moxidectin by dose and cohort (different disease severity).**

| Dose (mg) | PK Parameter[a] | | | | | | |
|---|---|---|---|---|---|---|---|
| | Cohort (n) | $C_{max}$ (ng/mL) | $T_{max}$ (h) | $AUC_{0-\infty}$ (days*ng/mL) | $T_{1/2}$ (days) | CL/F (L/day) | V/F (L) |
| 2 | 1 (10) | 16.3 (3.8) | 4.1 (2.4, 4.2) | 27.9 (7.9) | 17.6 (6.6) | 78.3 (27.7) | 1920 (781) |
| | 2 (11) | 16.2 (4.2) | 2.5 (2.0, 4.1) | 31.7 (17.5) | 21.8 (14.8) | 79.2 (37) | 2135 (968) |
| | 3 (12) | 17.3 (4.6) | 4.0 (2.1, 4.1) | 26.7 (10.7) | 21.9 (8.3) | 82.8 (22.1) | 2555 (1084) |
| 4 | 4 (11) | 33.4 (8.5) | 4.0 (2.4, 4.1) | 39.1 (12.7) | 13.9 (6.9) | 114.4 (42.8) | 2056 (750) |
| | 5 (11) | 34.5 (5.5) | 4.0 (2.1, 4.1) | 45.9 (11.8) | 17.4 (9.1) | 92.4 (22.6) | 2140 (794) |
| | 6 (12) | 35 (6.1) | 2.3 (2.0, 4.2) | 60 (26.9) | 21.7 (8.4) | 76.6 (25.3) | 2454 (1323) |
| 8 | 7 (12) | 55.7 (16) | 4.0 (2.2, 4.0) | 129 (98.6) | 35.8 (31) | 84.6 (37.6) | 3421 (2170) |
| | 8 (11) | 62.9 (24.9) | 4.0 (1.2, 4.0) | 99.5 (42.2) | 15.8 (7.7) | 89.6 (25.7) | 1887 (747) |
| | 9 (8) | 74.4 (13.5) | 4.0 (2.2, 4.0) | 111.8 (25.7) | 14.9 (4.7) | 75.5 (20.2) | 1655 (867) |

[a]Values are given as arithmetic means (SD) except for $T_{max}$, which shows median and (min, max).

in *O. volvulus* infected Africans is similar to that in healthy volunteers from Europe and the United States [21–26]. Drug concentrations rapidly reached a peak concentration in 4 hours. Following the peak concentration there is a very pronounced distribution phase where plasma levels decline rapidly by approximately 24 hours, followed by a prolonged elimination phase with a half-life of about 18 to 24 days. Cohort 7 was the only cohort with participants with measurable moxidectin concentrations at 6 and 12 months: one participant at 6 months, and one participant at 6 and 12 months. The very long elimination half-lives in these two

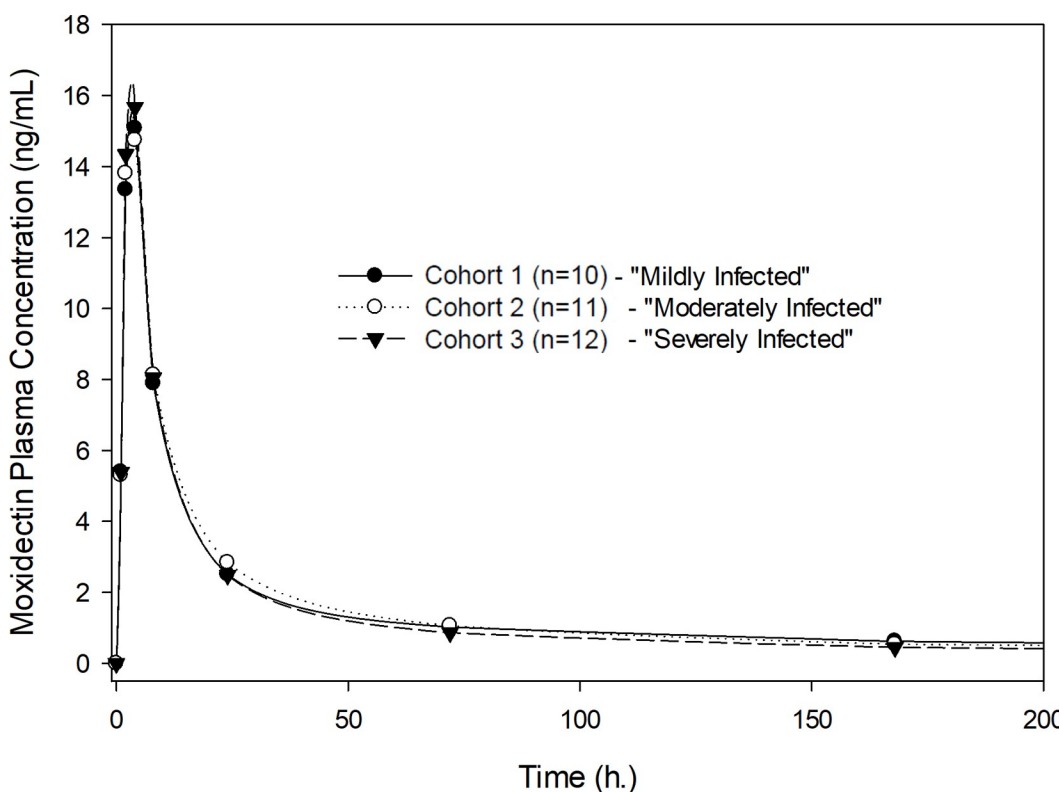

**Fig 2. Plasma concentrations of moxidectin after single oral administration of 2 mg by severity of infection, displayed as geometric means.**

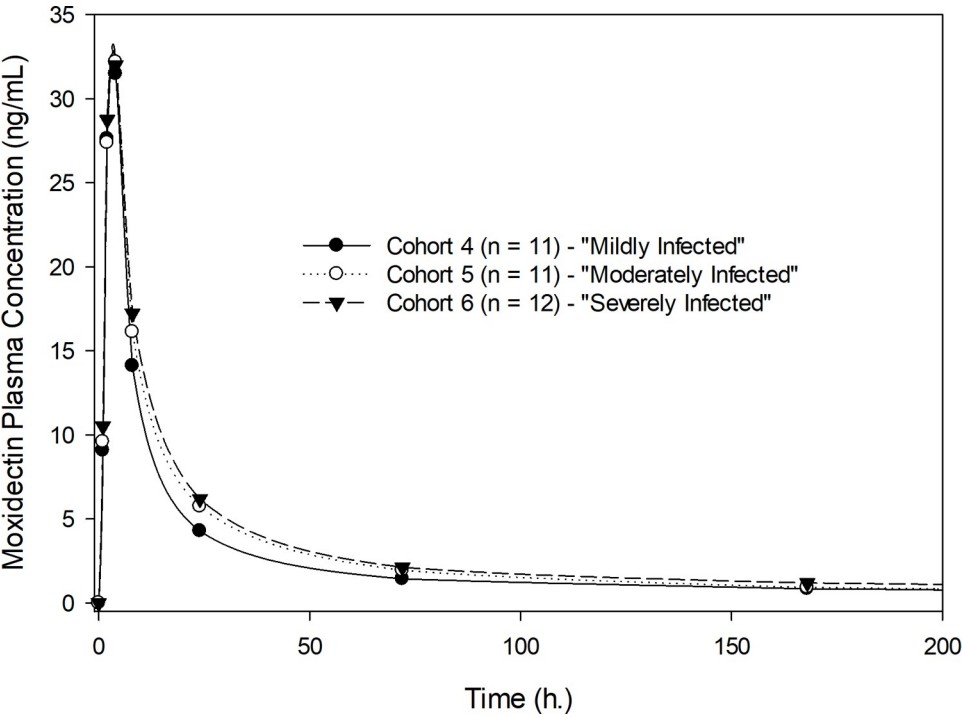

**Fig 3. Plasma concentrations of moxidectin after single oral administration of 4 mg by severity of infection, displayed as geometric means.**

participants resulted in high calculated parameters for V/F and thus a mean V/F for Cohort 7 much higher than the mean V/F for the other cohorts. No relationship with demographic characteristics could be established to account for this finding considering the relatively small number of participants in each cohort. Volume of distributions are similar in healthy

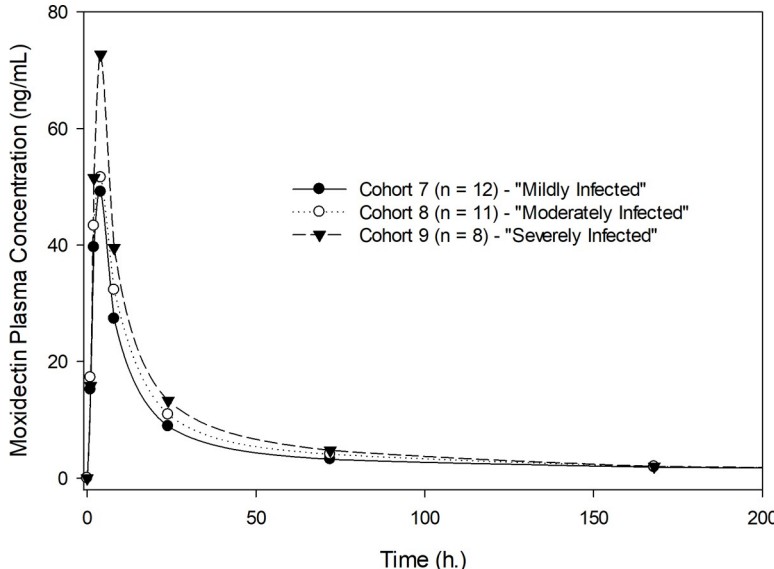

**Fig 4. Plasma concentrations of moxidectin after single oral administration of 8 mg by severity of infection, displayed as geometric means.**

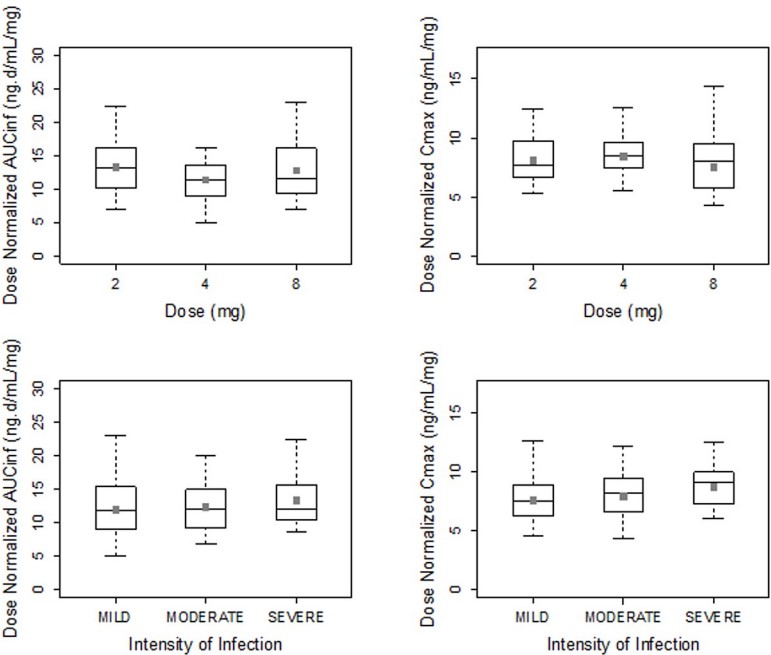

**Fig 5. Dose normalized plasma moxidectin exposures by dose and severity of infection.** Box plot provides median and 25%/75% quartiles with whiskers to the last point within 1.5 times interquartile range. Solid squares represent geometric mean.

volunteers and *O. volvulus* infected individuals, while apparent total clearance is 30% higher in *O. volvulus* infected individuals compared to healthy volunteers.

Moxidectin has predictable, dose-proportional pharmacokinetics in *O. volvulus* infected and healthy volunteers. Moxidectin exposures ($C_{max}$ and $AUC_{0-\infty}$) show a linear and dose-dependent increase with dose over the range of 2 to 8 mg in this study and 4 to 36 mg in fasting healthy volunteers [22,23]. A high-fat breakfast is associated with increased exposure and a delayed time to maximum concentration ($T_{max}$) [21]. Given the relatively wide margin of safety the increases are not clinically relevant and moxidectin can be dosed without regard to food, a conclusion reflected in the United States FDA approved labelling (https://www.accessdata.fda.gov/drugsatfda_docs/label/2018/210867lbl.pdf). Predictable pharmacokinetics and dosing without regard to food are helpful characteristics for a drug to be used in mass drug administration programs.

The drug concentration and length of residence time of a macrocyclic lactone in a target host tissue is a critical determinant of the efficacy of the drug against parasites in that tissue.

**Table 4. Summary of analysis of variance for dose-normalized exposure parameters of moxidectin.**

| Parameter | Source | Degree of freedom | Sum of squares | Mean square | F value | P-value |
|---|---|---|---|---|---|---|
| $AUC_{0-\infty}$ | Dose | 2 | 0.461 | 0.2306 | 1.536 | 0.221 |
| | Severity of infection | 2 | 0.265 | 0.1326 | 0.883 | 0.417 |
| | Interaction term | 4 | 0.886 | 0.2214 | 1.475 | 0.217 |
| | Residuals | 89 | 13.363 | 0.1502 | | |
| $C_{max}$ | Dose | 2 | 0.223 | 0.1117 | 1.754 | 0.179 |
| | Severity of infection | 2 | 0.280 | 0.1399 | 2.198 | 0.117 |
| | Interaction term | 4 | 0.238 | 0.0596 | 0.936 | 0.447 |
| | Residuals | 89 | 5.667 | 0.0637 | | |

Ivermectin has a small apparent volume of distribution and high oral clearance resulting in an elimination half-life of approximately 1 day and thus short residence time in the body [27]. On the other hand, moxidectin has a large apparent volume of distribution and low oral clearance, resulting in long terminal $T_{1/2}$ (mean values of 17.7–23.3 days in this study) in *O. volvulus* infected individuals.

The lack of noteworthy pharmacokinetic drug interactions is another important distinction between moxidectin and ivermectin. Moxidectin is a poor substrate for P-glycoprotein (P-gp) transporters, being mostly excreted via a P-gp-independent pathway into the intestine [28,29]. *In vitro* studies using mammalian liver microsomes suggested that moxidectin is a limited substrate of CYP-metabolism (cytochrome P450 3A and cytochrome P450 2B [30]), producing a small number of hydroxylated metabolites. There is no evidence for non-CYP-mediated metabolism. In humans, moxidectin is minimally metabolized within the body and does not affect the pharmacokinetics of midazolam, a sensitive CYP3A4 substrate [25]. These results lead to the conclusion that moxidectin exhibits no clinically relevant cytochrome P450-related drug–drug interactions. Moxidectin is excreted mostly unmetabolized in feces [31]. Renal elimination of intact drug is negligible. The percentage of the 8 mg moxidectin dose excreted into the breast milk of healthy female volunteers over 28 days was approximately 0.70% (approximately 0.056 mg) [26].

In conclusion, moxidectin has predictable, dose-proportional pharmacokinetics at clinically relevant doses. We found no relationship between severity of infection (mild, moderate or severe) and exposure parameters ($AUC_{0-\infty}$ and $C_{max}$), $T_{1/2}$ and $T_{max}$. Compared to the Phase I studies in fasting healthy volunteers, the pharmacokinetic parameters of terminal half-life, $AUC_{0-\infty}$, V/F and CL/F are consistent given expected biological variability and plasma sampling schedules. $T_{max}$ and $C_{max}$ obtained in the current study are also consistent with those reported in the Phase I studies [21–26].

## Acknowledgments

We recognize the contributions of the entire Onchocerciasis Clinical Research Centre, Hohoe staff (for further details see [18]). We are particularly grateful to all participants in the study for their co-operation.

The authors alone are responsible for the views expressed in this article and they do not necessarily represent the views, decisions or policies of the institutions with which they are affiliated.

## Author Contributions

**Conceptualization:** Kwablah Awadzi, Annette C. Kuesel, Janis Lazdins-Helds.

**Data curation:** Beesan Tan.

**Formal analysis:** Beesan Tan.

**Funding acquisition:** Kwablah Awadzi, Annette C. Kuesel, Janis Lazdins-Helds.

**Investigation:** Beesan Tan, Nicholas Opoku, Simon K. Attah, Kwablah Awadzi.

**Methodology:** Kwablah Awadzi, Lawrence Fleckenstein.

**Project administration:** Kwablah Awadzi, Annette C. Kuesel, Janis Lazdins-Helds, Mark Sullivan.

**Resources:** Kwablah Awadzi, Lawrence Fleckenstein.

**Software:** Beesan Tan.

**Supervision:** Lawrence Fleckenstein.

**Validation:** Lawrence Fleckenstein.

**Visualization:** Beesan Tan.

**Writing – original draft:** Beesan Tan.

**Writing – review & editing:** Beesan Tan, Nicholas Opoku, Simon K. Attah, Kwablah Awadzi, Annette C. Kuesel, Janis Lazdins-Helds, Craig Rayner, Victoria Ryg-Cornejo, Mark Sullivan, Lawrence Fleckenstein.

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
