## [Decision Letter · Decision Letter 0]

6 Jan 2022

Dear Prof. Fleckenstein,

Thank you very much for submitting your manuscript "Pharmacokinetics of oral moxidectin in individuals with Onchocerca volvulus infection" for consideration at PLOS Neglected Tropical Diseases. As with all papers reviewed by the journal, your manuscript was reviewed by members of the editorial board and by several independent reviewers. In light of the reviews (below this email), we would like to invite the resubmission of a significantly-revised version that takes into account the reviewers' comments. 

We cannot make any decision about publication until we have seen the revised manuscript and your response to the reviewers' comments. Your revised manuscript is also likely to be sent to reviewers for further evaluation.

Sincerely,

Jeremiah M. Ngondi, MB.ChB, MPhil, MFPH, Ph.D

Associate Editor

Timothy Geary

Deputy Editor

Reviewer's Responses to Questions

**Key Review Criteria Required for Acceptance?**

**Methods**

-Are the objectives of the study clearly articulated with a clear testable hypothesis stated?

-Is the study design appropriate to address the stated objectives?

-Is the population clearly described and appropriate for the hypothesis being tested?

-Is the sample size sufficient to ensure adequate power to address the hypothesis being tested?

-Were correct statistical analysis used to support conclusions?

-Are there concerns about ethical or regulatory requirements being met?

Reviewer #1: See general comments below. Hypotheses are not clearly articulated or supported.

Reviewer #2: Methods are clearly defined. some minor clarification for number or participants.

**Results**

-Does the analysis presented match the analysis plan?

-Are the results clearly and completely presented?

-Are the figures (Tables, Images) of sufficient quality for clarity?

Reviewer #1: Yes

Reviewer #2: Overall good - see comments

**Conclusions**

-Are the conclusions supported by the data presented?

-Are the limitations of analysis clearly described?

-Do the authors discuss how these data can be helpful to advance our understanding of the topic under study?

-Is public health relevance addressed?

Reviewer #1: Yes

Reviewer #2: (No Response)

**Editorial and Data Presentation Modifications?**

Reviewer #1: None

Reviewer #2: (No Response)

**Summary and General Comments**

Reviewer #1: Moxidectin is a macrocyclic lactone, like ivermectin, that was approved for human use against O. volvulus by the FDA in 2018. Moxidectin has an improved clinical profile relative to ivermectin, in that it suppresses microfilaria production more completely and for a longer period than ivermectin does, with a similar profile of adverse effects. The pharmacokinetics of moxidectin in uninfected people has been fully explored. In this paper, the authors have used serum samples collected as part of a clinical study of moxidectin efficacy published in 2014. The authors find that the pharmacokinetics of moxidectin in infected individuals did not differ from that of uninfected individuals.

Major comments:

1. The underlying hypothesis of this paper is that the pharmacokinetic profile of moxidectin would somehow be different in infected versus uninfected people. Since moxidectin is not metabolized, it is difficult for me to see why this might be the case. The authors should provide some additional rationale for undertaking the study. Are there any other known incidence of infection affecting the kinetics of a non-metabolizable drug, especially in a chronic infection like onchocerciasis?

2. Another underlying hypothesis here is that the AUC of moxidectin in the infected individuals would be greater than ivermectin. The authors did not address this at all. Given they presumably had access to the matched cohort that was given ivermectin in the clinical trial, this is an unfortunate omission. Adding these data would strengthen the manuscript considerably.

3. I originally had some serious concerns about the ethical underpinnings of the parental clinical trial. The individuals were chosen for participation in the study based on not having had any treatment for onchocerciasis for five years prior to the start of the study. As the clinical trial results were published in 2014 and Ghana reported treating all endemic villages as part of their elimination program by 1998, this seems to suggest the participants had been denied standard card leading up to the clinical trial. However, in the paper referenced by the authors that refers to the parental clinical trial, it was clear that the villages from which the participants were drawn were hypo-endemic and outside the blinding onchocerciasis zone, and thus had not been enrolled in CDTI before the start of the clinical trial. I think that this is very important information that needs to be included in this paper. The authors need to expand on their process for enrollment, in order to allay these concerns.

Reviewer #2: Major comments:

1. Line 107. It is noted that moxidectin is more lipophilic than ivermectin – are the authors suggesting an advantage for moxidectin? Please clarify the importance of including this statement.

2. Line 183-184. List the range for the standard curve.

3. Line 198. Was Clast always above the LLOQ? Were any concentrations below the LLOQ? If so this should be identified and how handled discussed.

4. Line 200. Both Cl and Volume of distribution should be labeled as “apparent”.

5. Line 212. The study referenced (#18) included 127 study participants who received moxidectin. Why were only 98 included in this report?

6. Line 216 – for the 2 mg dose 33 subjects were included (same in table 1). However in Figure 1 only 32 subjects are included – please clarify.

7. Table 2 – Tmax should be listed as median and range as it is considered a categorical variable and can only be either 3 or 4 as that is when sampling occurred.

8. Table 3- Volume of distribution is highly variable , 3421 L for cohort 7 vs 1655 L for cohort 9. This may warrant additional discussion, especially regarding demographics of cohorts (7,8,and 9).

9. The initial paragraph in the discussion really focuses on findings from reference 18. I would suggest either moving to the introduction or rearranging the discussion to focus on findings of the current study.

10. Line 339 – not clear that ivermectin has a small apparent volume of distribution, perhaps smaller than moxidectin. This should be clarified.

PLOS authors have the option to publish the peer review history of their article (what does this mean?). If published, this will include your full peer review and any attached files.

Reviewer #1: No

Reviewer #2: No
---

## [Editor Report · Decision Letter 1]

2 Mar 2022

Dear Prof. Fleckenstein,

We are pleased to inform you that your manuscript 'Pharmacokinetics of oral moxidectin in individuals with Onchocerca volvulus infection' has been provisionally accepted for publication in PLOS Neglected Tropical Diseases.

Best regards,

Jeremiah M. Ngondi, MB.ChB, MPhil, MFPH, Ph.D

Associate Editor

Timothy Geary

Deputy Editor

---

## [Editor Report · Acceptance letter]

22 Mar 2022

Dear Prof. Fleckenstein,

We are delighted to inform you that your manuscript, "Pharmacokinetics of oral moxidectin in individuals with Onchocerca volvulus infection," has been formally accepted for publication in PLOS Neglected Tropical Diseases.

Best regards,

Shaden Kamhawi

co-Editor-in-Chief

Paul Brindley

co-Editor-in-Chief
